# Predicting the animal hosts of coronaviruses from compositional biases of spike protein and whole genome sequences through machine learning

**Liam Brierley** *, **Anna Fowler**

Department of Health Data Science, University of Liverpool, Brownlow Street, Liverpool, United Kingdom

* liam.brierley@liverpool.ac.uk

## Abstract

The COVID-19 pandemic has demonstrated the serious potential for novel zoonotic coronaviruses to emerge and cause major outbreaks. The immediate animal origin of the causative virus, SARS-CoV-2, remains unknown, a notoriously challenging task for emerging disease investigations. Coevolution with hosts leads to specific evolutionary signatures within viral genomes that can inform likely animal origins. We obtained a set of 650 spike protein and 511 whole genome nucleotide sequences from 222 and 185 viruses belonging to the family *Coronaviridae*, respectively. We then trained random forest models independently on genome composition biases of spike protein and whole genome sequences, including dinucleotide and codon usage biases in order to predict animal host (of nine possible categories, including human). In hold-one-out cross-validation, predictive accuracy on unseen coronaviruses consistently reached ~73%, indicating evolutionary signal in spike proteins to be just as informative as whole genome sequences. However, different composition biases were informative in each case. Applying optimised random forest models to classify human sequences of MERS-CoV and SARS-CoV revealed evolutionary signatures consistent with their recognised intermediate hosts (camelids, carnivores), while human sequences of SARS-CoV-2 were predicted as having bat hosts (suborder Yinpterochiroptera), supporting bats as the suspected origins of the current pandemic. In addition to phylogeny, variation in genome composition can act as an informative approach to predict emerging virus traits as soon as sequences are available. More widely, this work demonstrates the potential in combining genetic resources with machine learning algorithms to address long-standing challenges in emerging infectious diseases.

## Author summary

New zoonotic viruses remain a major threat to global health and the COVID-19 pandemic has shown the specific potential of coronaviruses to cause widespread disease burden and economic damage. Tracing the origins of these zoonotic viruses is extremely challenging

**Data Availability Statement:** All raw data are available from the GenBank database (accession numbers provided in the supplementary data files). Processed data files and supporting code can be

obtained directly from https://github.com/lbrierley/cov_genome/.

**Funding:** LB acknowledges funding from a Medical Research Council Skills Development Fellowship award, grant number MR/T027355/1, https://mrc.ukri.org/ The funders had no role in study design, data collection and analysis, decision to publish, or preparation of the manuscript.

**Competing interests:** The authors have declared that no competing interests exist.

and usually requires substantial effort. However, there is potential to uncover which animals may be the host origin of viruses by using 'signatures' within viral genomes generated by long-term coevolution. We investigated this by calculating 116 genomic features of spike protein sequences and whole genome sequences from approximately 200 coronaviruses. We used a machine learning approach in random forests, training separate models to predict broad host type using genomic information from spike proteins or whole genomes. Models trained on spike proteins achieved similar performance to that of whole genomes, reiterating the importance of this protein for host-virus interactions and likelihood of cross-species transmission. When applied to SARS-CoV-2, the causative virus of COVID-19, model predictions suggested a bat origin, consistent with estimations elsewhere using more traditional phylogenetic analyses. This work demonstrates the potential of machine learning to infer the ecology of new zoonotic viruses directly from genetic sequences, giving a rapid methodology to assist in tracing the origins of outbreaks.

## Introduction

The ongoing COVID-19 pandemic remains a significant public health emergency. Since the first identified cases in China in December 2019, this outbreak of respiratory disease has developed into a global crisis, with over 100 million cases worldwide to date [1]. The causative virus was termed 'severe acute respiratory syndrome-related coronavirus 2' (SARS-CoV-2) [2] and is a previously unknown betacoronavirus that likely emerged through zoonotic transmission from contact with non-human animals [3,4]. However, the precise origins of the current pandemic remain inconclusive at present [5].

Two other betacoronaviruses have zoonotically emerged to cause significant human epidemics. Severe acute respiratory syndrome-related coronavirus (SARS-CoV) emerged in China in 2002 via an intermediate host of masked palm civets (*Paguma larvata*) in live animal markets [6,7], and Middle East respiratory syndrome related coronavirus (MERS-CoV) emerged in Saudi Arabia in 2012 via an intermediate host of dromedary camels [8,9], with considerable evidence that both originated in bats [10–13]. Four additional coronaviruses are known to be endemic within humans, causing mild common cold-like illness (*Alphacoronavirus*: Human coronaviruses 229E and NL63; *Betacoronavirus*: Human coronaviruses HKU1 and OC43).

All viruses in the family *Coronaviridae* feature similar structural proteins, including a spike glycoprotein on the outer viral surface. This protein allows entry to host cells via attachment to cell receptors at the receptor binding domain (RBD) region and subsequent fusion with cell membranes. As such, this protein exhibits high variation even between closely related coronaviruses that may correspond to profound differences in receptor usage and/or different tissue and host preferences [14]. While SARS-CoV-2 shows high genetic similarity to bat coronaviruses, particularly bat coronavirus RaTG13 (matching 96% sequence identity) [4], its spike protein instead exhibits differences among key amino acid residues of the RBD [3,15]. Based on this, SARS-CoV-2 is predicted through structural [15,16] and in vitro experimental models [17,18] to have highly efficient binding to the human angiotensin-converting enzyme 2 (ACE2) receptor, a feature that has likely contributed to its efficient human-to-human transmissibility. Several methods, e.g., ENC analysis of codon usage for SARS-CoV-2 [19] and positive selection analysis for MERS-CoV [20,21] have demonstrated that the spike proteins of these coronaviruses experienced substantial selection pressure during emergence compared to other genomic regions. Therefore, as a key molecular determinant of host range [14,22],

evolutionary adaptation of coronavirus spike proteins represent a key opportunity to further understand their host range constraints.

Beyond selection acting at specific loci, viral adaptation can also manifest through broad-scale genomic signatures. Viruses exhibit biased genome composition, for example, in non-uniform use of synonymous codons [23]. Furthermore, coevolution within different hosts may indirectly lead to selection for particular compositions, as reported for nucleotide and dinucleotide usage within avian and human influenzaviruses [24,25] and codon pair usage of arboviruses within their insect vectors and mammalian hosts [26]. The *Coronaviridae* are no exception—different coronaviruses (including SARS-CoV-2) vary in their genome composition, with particularly complex patterns of codon usage in spike protein coding sequences [27], which could potentially contain important evolutionary signal regarding host origin.

Machine learning has recently gained substantial attention as a methodology in comparative modelling of emerging diseases. These methods are capable of decomposing signal in high-dimensional genomic information (a limitation of regression frameworks) without the need for sequence alignment. Genomic machine learning analyses have demonstrated the ability to not only classify viruses from recurring viral genome motifs [28], but also classify their broad host origins [29–32]. Specifically considering coronaviruses, support vector machines and random forests have been trained on various genomic features to predict host group, including nucleotide and dinucleotide biases [33], amino acid composition [34] or sequence k-mers [35]. However, previous model predictions are mostly concentrated upon bats or humans, and few analyses explicitly address the spike protein (but see [35]). The exact potential of genome composition to predict host origin therefore remains unclear.

We aimed to use machine learning to understand how the complex genomic signatures of coronaviruses might predict their hosts and determine the importance of such signature in the spike protein. Specifically, we trained random forest models on compositional biases for spike protein and whole genome nucleotide sequences and compared their performance. A limitation of these approaches is that model predictions can be a) strongly influenced by the inherent viral sampling biases in available data [36] and b) reflect evolutionary relatedness of viral lineages rather than true host-associated signal [37]. Therefore, we conduct data thinning of over-represented viruses before training models and control for similarity between related sequences by holding out entire species during cross-validation. We demonstrate the use of machine learning as a reliable method to estimate host origins of future novel coronaviruses in humans and livestock.

## Materials and methods

### Data extraction and processing

Spike protein or whole genome sequence data for coronaviruses were identified within Gen-Bank, using search terms

$txid\#\#\#[Organism : noexp]$ AND $(spike[Title]$ OR ″S gene″$[Title]$ OR ″S protein″$[Title]$ OR″S glycoprotein″$[Title]$ OR ″S1 gene″$[Title]$ OR ″S1 protein″$[Title]$ OR″S1 glycoprotein″$[Title]$ OR peplomer$[Title]$ OR peplomeric$[Title]$ OR peplomers$[Title]$ OR″complete genome″$[Title])$ NOT $(patent[Title]$ OR vaccine OR artificial OR construct OR recombinant$[Title])$

where successive searches were conducted replacing ### with taxonomic identifiers for each species and unranked sub-species belonging to the family *Coronaviridae* within the NCBI taxonomy database [38] (n = 1585 taxonomic ids total). Matching sequences were then extracted and further filtered to exclude incomplete or truncated sequences based on a) metadata labels

and b) length restrictions, discarding any spike protein sequences < 2 kilobases (kb) and any whole genome sequences outside a range of 20kb– 32kb. We accepted both spike protein coding sequences within whole genome sequences and standalone complete spike protein sequences, excluding those only covering individual S1 or S2 subunits. All sequence data searching, filtering and extraction was conducted with R package 'rentrez'v1.2.2 [39] (see also [40]).

## Host classification

For each spike protein or whole genome sequence, host names were extracted from the host organism metadata field before being resolved to the standard NCBI taxonomy using the R package 'taxizedb'v0.1.9.93 [41] (see also [40]). Host names were automatically resolved to the highest taxonomic resolution possible and any unmatched host names were resolved manually, discarding sequences with missing/unresolvable names.

We then constructed a new variable broadly describing host category of each sequence, defined at various taxonomic levels: human (species *Homo sapiens*), camelid (family *Camelidae*), swine (family *Suidae*), carnivore (order *Carnivora*), rodent (order *Rodentia*), and bird (class *Aves*). Following a previous analysis [29], we included two categories to represent bats (order *Chiroptera*): suborder Yinpterochiroptera (families *Craseonycteridae*, *Hipposideridae*, *Megadermatidae*, *Pteropodidae*, *Rhinolophidae*, and *Rhinopomatidae*) and suborder Yangochiroptera (all other families), based on their evolutionary divergence [42] and differences in ecology and host-virus relationships [43,44]. Sequences not conforming to any of the above host categories were excluded from further analysis.

## Genomic feature calculation

We then calculated several features describing genome composition biases of each spike protein and whole genome coding sequence at nucleotide, dinucleotide or codon level. Firstly, nucleotide biases were calculated as simple proportion of A, C, G or U content. Dinucleotide biases were calculated as the ratio of observed dinucleotide frequency to expected based on nucleotide frequency, following [29]:

$$\frac{\frac{d_{xy}}{D}}{\left(\frac{n_x}{N} \cdot \frac{n_y}{N}\right)}$$

where $d_{xy}$ denotes frequency of dinucleotide $xy$, $n_x$ and $n_y$ denote frequency of individual nucleotides $x$ and $y$, and $D$ denotes total dinucleotides and $N$ total nucleotides for length of the given sequence. Biases were calculated separately for each dinucleotide at each position within codon reading frames (i.e., at positions 1–2, 2–3 or 3–1) as dinucleotides spanning adjacent codons may be subject to more extreme biases [45,46]. Finally, Relative Synonymous Codon Usage (RSCU) was also calculated for each codon including stop codons, following [47]:

$$\frac{c_{ij}}{\frac{1}{n_i} \cdot \sum_{j}^{n_i} c_{ij}}$$

where $n_i$ denotes number of codons synonymous for amino acid $i$ and $c_{ij}$ denotes frequency of $j^{th}$ codon encoding for such amino acid. In total, this gave 116 genomic features for use in predictive models (4 nucleotide biases, 16 * 3 dinucleotide biases, 64 codon biases). The Effective Number of codons (ENc) [48] was also calculated for each sequence as a summative metric of magnitude of codon bias. All calculations for whole genomes considered nucleotide sequences as-is rather than as-read; sequence strings duplicated by frameshifting among the ORF1a and

ORF1b replicase protein were discarded to avoid disproportionate weighting in modelling analyses. Nucleotide and dinucleotide frequencies were obtained using R package 'Biostrings', v2.56.0 [49], and codon frequencies and ENC values were obtained using R package 'cordon', v.1.6.0 [50].

To assess over- or underrepresentation of codons among spike protein and whole genome datasets, we conducted one-sample t-tests comparing RSCU for each codon to a null value of 1 (excluding stop codons and codons with no other synonymous codons), applying Bonferroni corrections for multiple testing. Clustering of RSCU within-genus was assessed by constructing heatmaps using package 'gplots', v3.0.3 [51] before extracting associated dendrogram clusters and calculating Normalised Mutual Information (NMI) with genus, a measure of co-occurrence scaled between 0 (no mutual information) and 1 (perfect information) (see [52]). NMI was calculated using function 'NMI()'in package 'aricode', v.1.0.0 [53], cutting dendrograms into k = 10 clusters and excluding viruses unclassified into a genus.

## Machine learning analysis

To quantify the potential for genome composition biases to predict coronavirus host category, we used random forests, an ensemble machine learning approach that aggregates over a large number of individual classification tree models [54]. We selected random forests as a method that natively handles multiclass classification problems, intuitively handles interactions between predictor features, and can provide interpretability in feature-outcome explanatory relationships (see below) [55].

These predictive modelling methods are often sensitive to training data composition, a potential problem given the heavily biased patterns of viral sampling [36]. Several viruses appeared highly overrepresented in extracted data, for example, 1802 spike protein and 555 whole genome sequences were available for porcine epidemic diarrhea virus (S1 Table), compared to the overall median of 1 sequence per virus in both cases. As viral sequences (and therefore, genome composition) are expected to be highly similar within-species or subspecies, we therefore conducted a heuristic data thinning procedure prior to modelling, sampling a maximum of twenty sequences per host category per virus, thinning for approximately 5% of viruses (S1 and S2 Tables).

Machine learning methods can also be sensitive to class imbalances in outcomes [56] and slight imbalances between host categories remained after data thinning (S2 Table). Preliminary models suggested predictive performance was robust to this imbalance, being comparable or decreasing when alternative resampling methods were applied (S1 Methods and S3 and S4 Tables).

Zoonotic and epizootic coronaviruses are unlikely to have experienced substantial coevolution within the novel host following cross-species transmission. Instead, their evolutionary signature in genome composition is much more likely to reflect the original or donor host. Therefore, zoonotic or epizootic coronavirus sequences sampled from novel hosts (i.e., SARS-CoV, SARS-CoV-2, MERS-CoV in humans, totaling m = 9 taxonomic identifiers, full list in S1–S4 Data; swine acute diarrhea syndrome coronavirus in swine) were held out from model training. Of these, we retained the zoonotic sequences as a prediction set of interest. By generating predictions for zoonotic viruses from sequences alone, we aimed to investigate model utility when presented with the scenario of a newly identified emerging coronavirus. We excluded human enteric coronavirus from all analyses as the zoonotic potential of this virus remains unclear.

As genome composition is highly homogenous for closely related virus sequences, we established training data for random forest models using an outer loop of hold-one-out validation

applied to coronaviruses (S1A Fig), i.e., rather than exclude randomly sampled sequences as a test set, we constructed and validated random forests excluding all sequences from a given virus species or unranked subspecies in each instance. This allows host predictions for unseen (i.e., novel) viruses based on values of compositional features, rather than indirectly predicting host by the proxy of viral similarity. For each outer loop, model parameters were then optimized within an inner loop of 10-fold cross-validation, dividing the data into ten folds, training models using a grid search of parameter combinations on nine and identifying the best parameters when applied to the tenth fold in each case (S1B Fig). The parameter set yielding the highest prediction accuracy on validation folds was retained (S2 Fig).

Model performance was then assessed by applying each random forest to its respective held-out coronavirus sequences as a test set. Probabilities of host categories were obtained by dropping sequences down each individual tree model within the random forest and averaging host category prevalence of resulting terminal nodes [57]. Similar predictions were generated for each zoonotic coronavirus sequence by averaging probabilities across all random forests to generate grand mean probabilities (S1A Fig). To investigate explanatory relationships between genomic biases and hosts of coronaviruses, variable importance (calculated as relative mean decrease in Gini impurity) and partial dependence (calculated as marginal probability of each host category) associated with each genomic feature were averaged across all random forests.

All random forests were constructed using 1000 trees and implemented using R package 'ranger', v0.12.1 [58]. All analyses were initiated from a fixed random seed chosen a priori. Model performance was insensitive to choice of random seed (S5 Table). All data processing and modelling were conducted within R v4.0.0 [59]. Supporting data and code are available at https://github.com/lbrierley/cov_genome.

## Results

### Genome composition across the *Coronaviridae*

In total, we identified n = 3595 nucleotide sequences for coronavirus spike proteins and n = 1815 whole genome sequences that met inclusion criteria. These were thinned to n = 650 spike protein sequences from m = 222 coronaviruses and n = 511 whole genome sequences from m = 185 coronaviruses for use in further analysis (S1 Fig and S2 Table), spanning 40 identified host genera and 58 identified host species (S1 and S2 Data).

Broadly consistent genome composition biases were observed across the diversity of all coronavirus sequences examined. For both spike protein and whole genome sequences, all A- and U-ending codons were significantly overrepresented (one-sample t-tests of RSCU, Bonferroni-corrected p < 0.001) except AUA, GCA, CUA, GGA, and GUA, and all C- and G-ending codons were underrepresented (Bonferroni-corrected p < 0.001) except UUG (and AGG in the case of spike proteins only) (Figs 1 and 2). Hierarchical clustering based on RSCU values suggested codon usage was less distinct between genera within spike protein sequences (Normalised Mutual Information value = 0.268; Fig 1) than within whole genomes (NMI = 0.428; Fig 2). However, clear separation of deltacoronaviruses was observed for both cases, as these appeared to have less extreme biases in codon usage. This was confirmed by ENc calculation; deltacoronaviruses had higher ENc values than other coronavirus genera (Table 1). Considering dinucleotide biases, compositional bias was typically more extreme for dinucleotides spanning adjacent codons, i.e., position 3–1 (S3 Fig), and the characteristic coronavirus CpG suppression was also observed.

### Host predictions of random forest models

Random forest models trained on nucleotide, dinucleotide and codon bias features of spike protein sequences predicted coronavirus hosts with 73.5% accuracy during hold-one-out

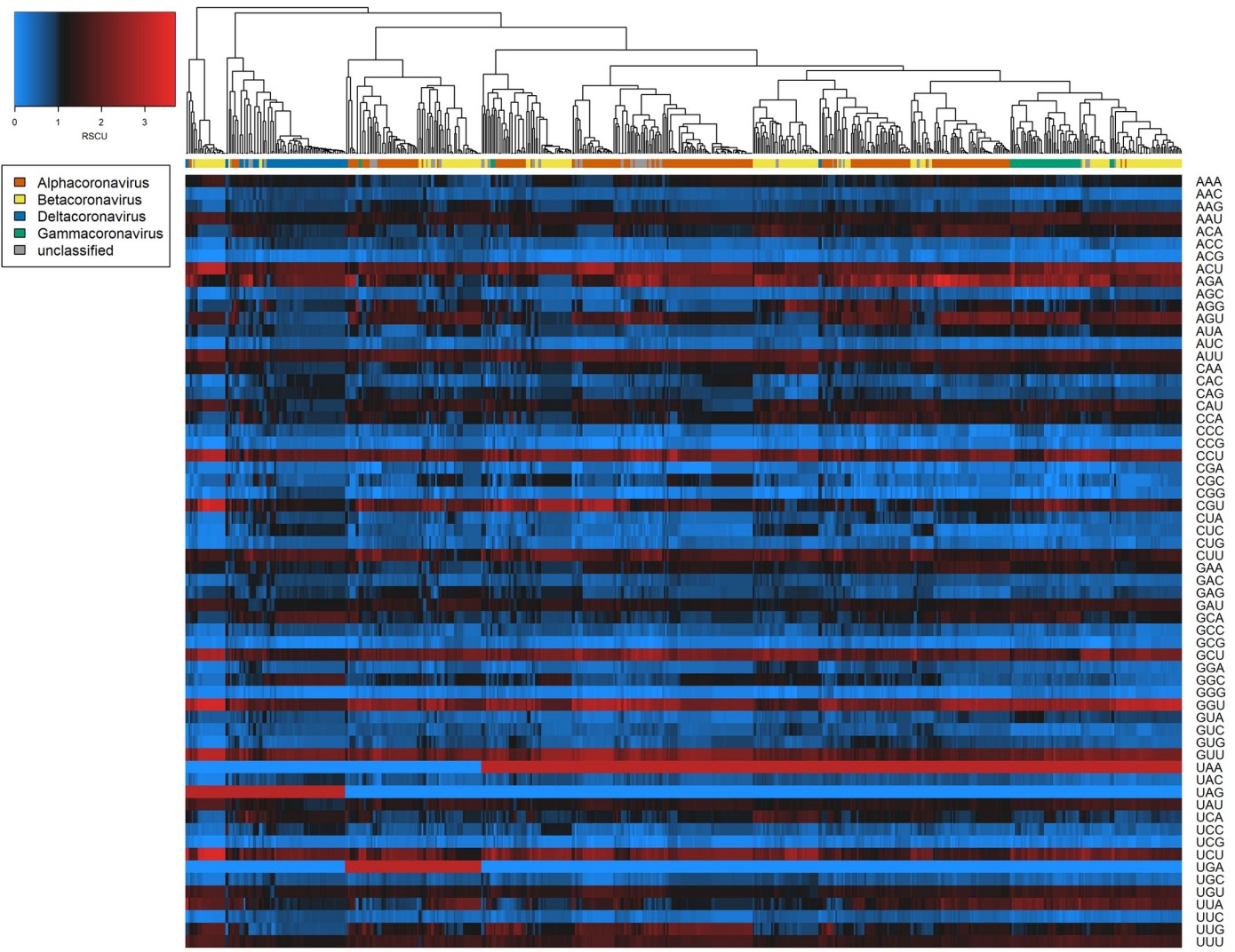

**Fig 1. Codon biases (RSCU) across coronavirus spike protein sequences examined.** Heatmaps of coronavirus codon usage bias (RSCU) associated with each codon in each spike protein sequence (n = 650). Main colour scale denotes RSCU value, a null value of 1 (black) indicating no difference in codon usage from expectation, with blue and red representing under- or overrepresentation respectively. Dendrogram colour bar denotes taxonomic genus.

cross-validation (Table 2). Genome composition of spike proteins appeared just as informative as whole genomes despite being much smaller in sequence length, as both models achieved very similar performance in all diagnostic measures (Table 2).

Patterns of host-specific predictive performance were evident during hold-one-out cross-validation. Random forests trained on both spike protein and whole genome sequence compositional features most easily distinguished bird, carnivore and rodent host categories (Fig 3 and S6 and S7 Tables). Less powerful predictive performance was obtained for livestock (i.e., swine and camelid) host categories with these sequences often predicted as having bat (suborder Yangochiroptera) hosts, including all MERS-CoV sequences sampled from camels.

Human host origins appeared particularly difficult to characterise, with model-predicted hosts appearing more uncertain using spike protein features than whole genome features (Fig 3); while human coronaviruses HKU1 and NL63 were more confidently correctly classified based on whole genomes, human coronaviruses OC43 and 229E were more confidently

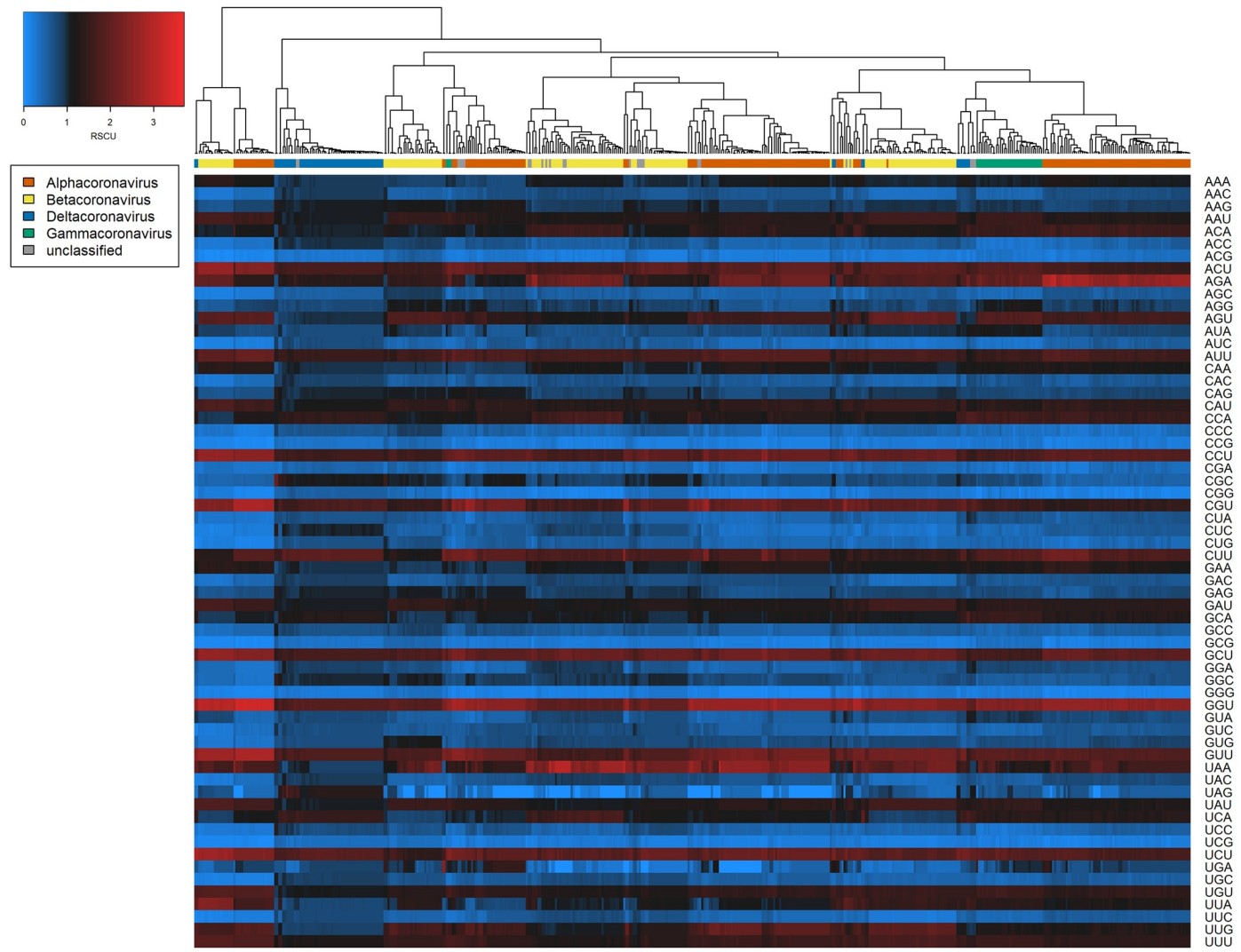

**Fig 2. Codon biases (RSCU) across coronavirus whole genome sequences examined.** Heatmaps of coronavirus codon usage bias (RSCU) associated with each codon in each whole genome sequence (n = 511). Main colour scale denotes RSCU value, a null value of 1 (black) indicating no difference in codon usage from expectation, with blue and red representing under- or overrepresentation respectively. Dendrogram colour bar denotes taxonomic genus.

**Table 1. ENc values across coronavirus genera.** Effective Number of Codons (ENc) for coronaviruses, stratified by genus and genome sequence type. ENc values are calculated as grand means, i.e., mean ENc was calculated per coronavirus by averaging sequences before means of means were calculated per genus by averaging coronaviruses. SD denotes standard deviation.

| | Spike protein sequences | | Whole genome sequences | |
|---|---|---|---|---|
| **Genus** | **Mean** | **SD** | **Mean** | **SD** |
| *Alphacoronavirus* | 48.65 | 3.12 | 45.02 | 3.00 |
| *Betacoronavirus* | 47.77 | 5.04 | 47.03 | 4.79 |
| *Gammacoronavirus* | 46.36 | 1.83 | 46.11 | 0.74 |
| *Deltacoronavirus* | 54.79 | 3.43 | 51.75 | 3.22 |
| (unclassified) | 47.89 | 3.41 | 48.56 | 3.04 |
| **total** | 48.77 | 4.36 | 46.81 | 4.22 |

**Table 2. Predictive performance of random forest models.** Model diagnostics describing overall performance when applied to predict host category of held-out coronaviruses not used for model training. CI denotes confidence interval, Kappa denotes Cohen's Kappa statistic, mAUC denotes multiclass area-under-curve statistic, and F1macro denotes F1 score calculated using macro-averaging (performance on each host category weighted equally).

| Predictor features | Accuracy (95% CI) | Kappa | mAUC | $F1_{macro}$ |
| --- | --- | --- | --- | --- |
| Spike protein | 0.735 (0.700, 0.769) | 0.696 | 0.898 | 0.757 |
| Whole genome | 0.728 (0.687, 0.766) | 0.688 | 0.902 | 0.758 |

misclassified as having camelid or Yinpterochiroptera hosts. The reciprocal also only occurred using whole genomes, i.e., several camel coronaviruses were predicted to have human hosts.

We then applied random forests to those sequences of zoonotic viruses sampled from humans and excluded from model training: SARS-CoV, SARS-CoV-2, and MERS-CoV (S1 Fig and S3 and S4 Data). As these viruses have experienced little coevolution following zoonotic spillover, their genome composition signal likely gives indications about their ultimate or proximate animal host origins. MERS-CoV was overwhelmingly predicted to have camelid hosts (Fig 4) and SARS-CoV was predicted with less certainty as having carnivore hosts, consistent with the respective known intermediate hosts of camels and palm civets (order *Carnivora*). Contrastingly, SARS-CoV-2 was predicted mostly strongly to have a bat (suborder

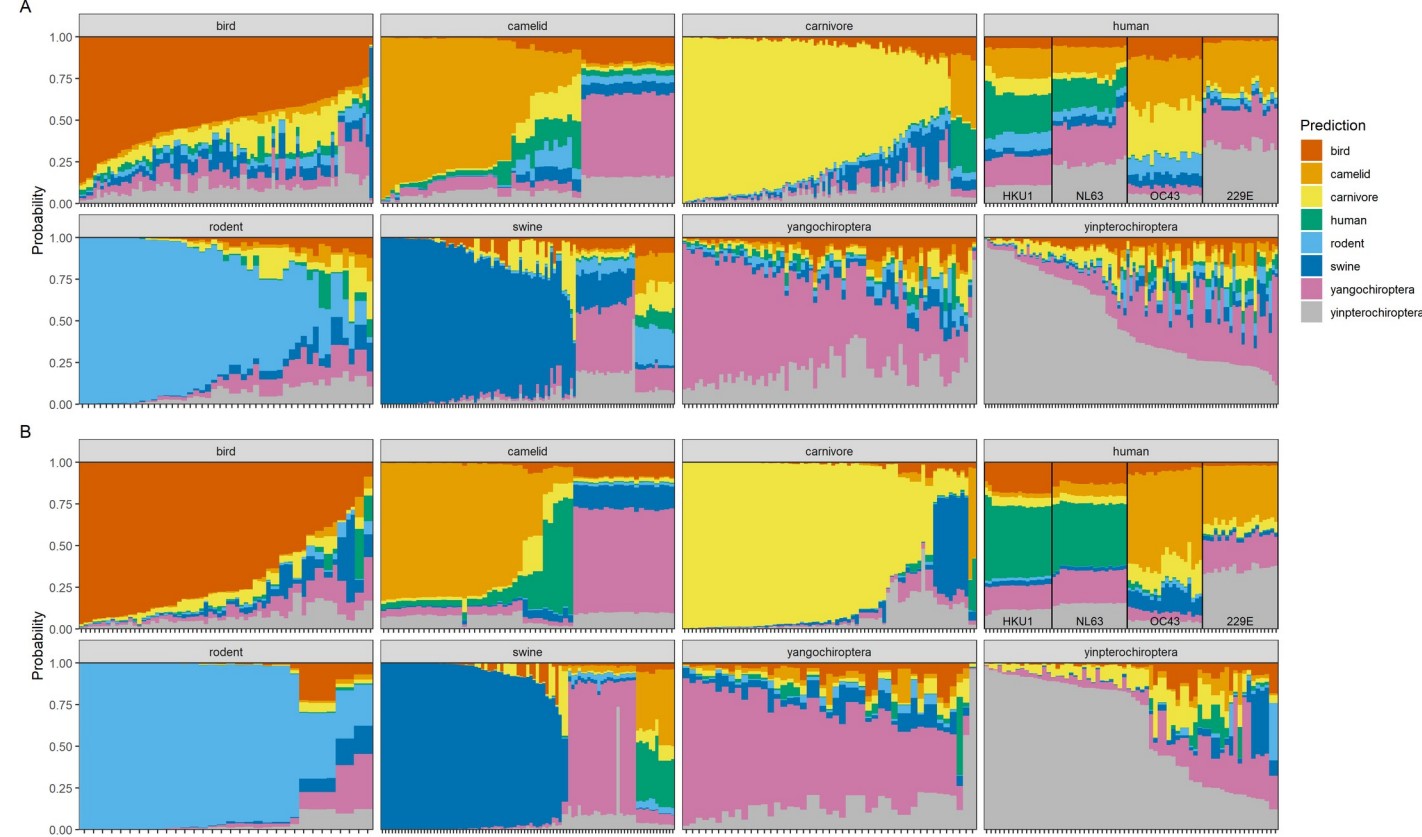

**Fig 3. Random forest host predictions based on coronavirus genome composition.** Stacked bar plots of predicted probabilities of each host category for coronavirus sequences. Predictions were obtained from ensemble random forest models trained on A) spike protein and B) whole genome composition features. Panels depict sequences from each metadata-derived host category and colour coding denotes model-predicted host category. Stacks represent individual coronavirus sequences, ordered from largest to smallest probability of the correct host, i.e., greater panel area matching the correct host category indicates better overall model performance. Non-zoonotic coronavirus sequences originating from humans (human coronaviruses HKU1, NL63, OC43, 229E) are labelled for clarity. Versions stratified by genera and species are provided as S4 and S5 Figs.

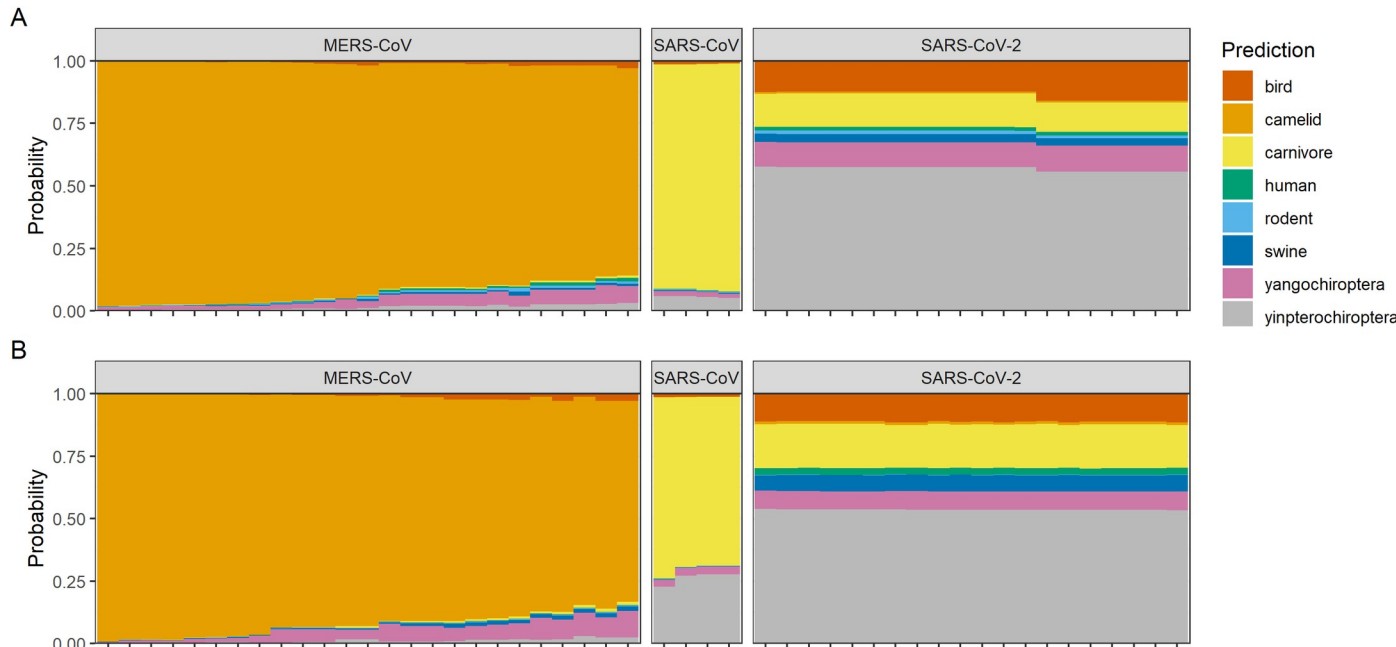

**Fig 4. Random forest predictions based on zoonotic coronavirus genome composition.** Stacked bar plots of predicted probabilities of each host category for zoonotic coronavirus sequences sampled from humans. Predictions were obtained from ensemble random forest models trained on A) spike protein and B) whole genome composition features. Colour coding denotes model-predicted host category. Stacks represent individual coronavirus sequences.

Yinpterochiroptera) host. Host predictions for zoonotic viruses were consistent between models using spike protein and whole genome features (Fig 4).

## Variable importance of random forest models

The most informative genomic features towards predicting coronavirus hosts were a mixture of dinucleotide and codon biases (Fig 5), with dinucleotide biases appearing slightly more informative for spike protein sequences and codon biases appearing slightly more informative for whole genome sequences. However, predictive power of individual genomic features did not hold between spike protein and whole genome sequences; only weak correlation was observed between ranked variable importance from both analyses (Spearman's rank, $\rho$ = 0.191, p = 0.042) (Figs 5, S6 and S7). Partial dependence plots suggested the strongest individual discriminating feature to be GG dinucleotides at positions 1–2; an overrepresentation of this dinucleotide within the spike protein sequence clearly distinguished bird hosts from mammalian hosts (S6 Fig), consistent with the greatest predictive performance for bird coronaviruses (S6 Table).

## Discussion

We observe biased dinucleotide and nucleotide usage across the family *Coronaviridae*, and demonstrate that these genome composition biases contain sufficient evolutionary signal such that they can predict animal host origin. We show that training random forests on these features of spike proteins is equally as informative as using whole genome sequences in predicting hosts of novel (i.e., unseen during model training) coronaviruses, with bird, carnivore and rodent viruses having the highest prediction accuracy. When applied to human coronavirus sequences from previous epidemics (SARS-CoV, MERS-CoV), random forest model

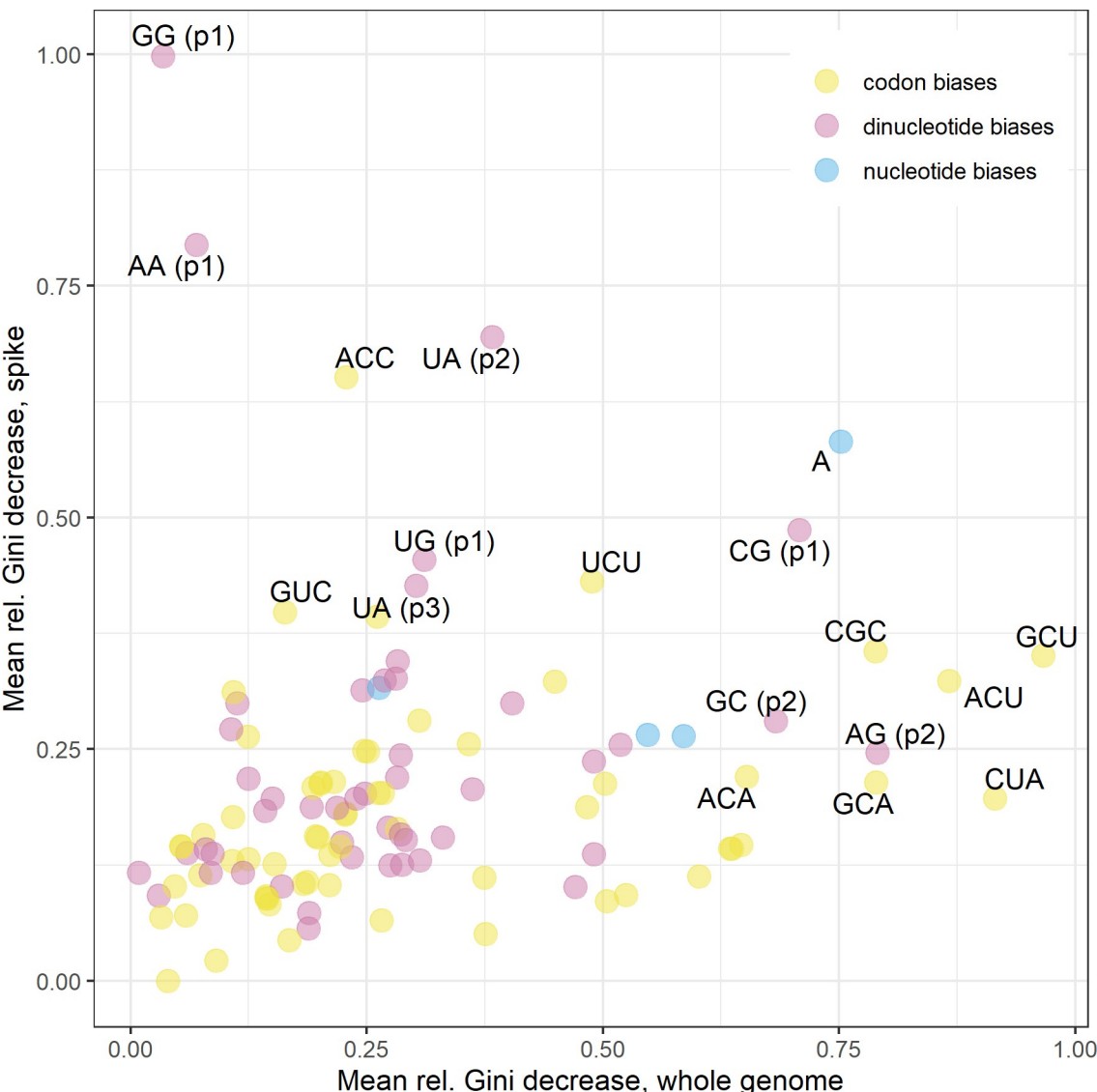

**Fig 5. Variable importance of genomic features.** Variable importance of genome composition features in ensemble random forest models predicting coronavirus host category from whole genome sequences (x axis) and spike protein sequences (y axis), with labelling of top ten most informative features from both analyses. Points denote mean values of relative decrease in Gini impurity associated with each feature across A) m = 222 and B) m = 185 random forests during hold-one-out cross-validation. Colour key denotes genomic feature type.

predictions consistently represented the intermediate hosts. In the case of SARS-CoV-2, where the exact transmission pathway remains unknown, models predicted sequences to have a bat host (suborder Yinpterochiroptera).

## Variability in genome composition

Among our dataset of 222 coronaviruses, we observed A- and U-ending codons to be overrepresented and C- and G-ending codons to be underrepresented (Figs 1 and 2), a commonly noted trait in other studies [19,60]. Elsewhere, CpG dinucleotide bias has been proposed as a specific determinant of host (and tissue) range of coronaviruses [61], on the basis that CpG dinucleotides are targeted by zinc finger antiviral proteins and their suppression is therefore

linked to immune evasion. We observed consistent CpG suppression (S3 Fig) and CG dinucleotides (positions 1–2) ranked 6[th] and 8[th] in feature importance for spike proteins and whole genomes (Fig 5), respectively.

However, spike proteins display different patterns of codon usage from other viral proteins [27], reflected in the lack of correlation between genome composition feature importance in random forests trained on spike proteins versus whole genomes (Figs 5, S6 and S7). This indicates spike proteins contain complex evolutionary signatures reflecting their distinct role in host-virus coevolution, supporting our approach in using many features beyond single dinucleotides [62]. The same evolutionary signatures would not be expected of proteins not involved in direct molecular interactions with hosts; in a comparative model trained on genome composition of the envelope protein (which has codon usage consistent with mutational bias rather than selective pressure [19]) predictive performance was reduced (S8 Table). Further work is needed to address the individual products of the ORF1ab polyprotein as an additional area of the genome under selective pressure for codon usage [19].

### Model predictions of human coronaviruses

During model validation, human hosts appeared more challenging to correctly predict than other host types. Although the endemic human coronaviruses are common, they are also thought to have their ultimate evolutionary origins within non-human animals [11], which may explain this difficulty. In particular, human coronaviruses NL63 and HKU1 were more consistently predicted as having human hosts than human coronaviruses OC43 and 229E, especially when using whole genome sequences (Fig 3). Human coronavirus NL63 is estimated to have a more ancient common ancestor with bat coronaviruses than 229E [63,64], implying longer coevolutionary history within human hosts has resulted in a more consistently identifiable genomic signature.

Although several mutations of SARS-CoV-2 are becoming fixed in the population, e.g. D614G in the spike protein [65], the virus has experienced only weak purifying selection [66] and sequences remain extremely similar over the course of the pandemic. As such, our approach cannot identify host adaptation "in real-time"; rather, we examine variation generated over much longer macroevolutionary histories.

Instead, we would expect viruses that have transmitted cross-species more recently to retain the genome composition signature of their previous hosts, having experienced little coevolution within the novel host. Applying our finalised models to zoonotic human virus sequences may therefore give an indication of their proximate or ultimate animal host origin (Fig 4).

Human sequences of SARS-CoV were predicted to have a carnivore host, consistent with the known intermediate host of palm civets (*Paguma larvata)*. Much previous work has shown human and civet SARS-CoV sequences to have high similarity, with adaptive mutations concentrated within the spike protein (specifically, the receptor binding domain) [6,7,22], which may explain the stronger prediction of carnivore hosts when using spike proteins than whole genome sequences (mean (SD) predicted probabilities: 0.902 (0.007) vs 0.688 (0.024)). Similarly, human sequences of MERS-CoV were strongly predicted as having camelid hosts, consistent with the intermediate host of dromedary camels. The detection of evolutionary signatures corresponding to these intermediate hosts implies that these coronaviruses circulated in those hosts for sufficient time for coevolution to shape genome composition before cross-species transmission to humans. For MERS-CoV, camel infections have been recognised as far back as at least the 1980s [67,68].

The origins of SARS-CoV-2 have been heavily speculated upon since its discovery, though there remains no compelling evidence towards the animal source of the first human infections.

Our random forest models trained on genome composition of both spike protein and whole genome sequences predicted SARS-CoV-2 as having a bat host (suborder Yinpterochiroptera). Alignment-based and phylogenetic approaches showed the most closely related virus to be bat coronavirus RaTG13, a virus sampled from a horseshoe bat (*Rhinolophus affinis*) belonging to this suborder [4], and more widely, the *Rhinolophidae* family are the most likely ancestral hosts of the *Sarbecovirus* genus [69].

While our model predictions support bats as the ultimate origin of SARS-CoV-2, the involvement of intermediate hosts remains unclear. Although the Malayan pangolin (*Manis javanica*) was proposed early in the pandemic [70,71], recent analyses have argued there is absence of evidence for this [72,73]. The methods used here are unable to identify intermediate hosts without sufficient sequence availability, and lack of such from pangolins (order *Pholidota*) preclude us from directly testing this hypothesis. However, selection analyses indicate SARS-CoV-2 could reasonably have exhibited efficient human infectivity and human-to-human transmissibility following direct transmission from bats [66,73], i.e., without strict need for prolonged selection within an intermediate host.

## Future directions

A natural comparison to these methods is phylogenetic analyses, which can estimate traits such as host type from reconstructing viral ancestry based on sequence similarity. There is challenging confounding between molecular characteristics and sequence similarity, i.e., variation in genome composition may actually be predictive of viral lineage rather than host type [37], essentially acting as a proxy for phylogenetic relatedness. To separate these signals, viral similarity needs to be considered in model construction [31]; by using a cross-validation procedure holding out entire coronavirus species or sub-species, we attempt to distinguish genomic signatures arising from convergent evolution within specific hosts, rather than from viral similarity. A more generalised scope of study across multiple viral families would allow holdout of entire families during cross-validation [31], removing any further phylogenetic proxy effects.

Additional challenges are created by the unavoidable, systematic gaps in sampling coverage. For example, disproportionate sampling to identify viruses in wildlife similar to those already known to affect humans or livestock may introduce bias to predictive models [36]. Although we address this by data thinning, our model predictions, particularly for zoonotic coronaviruses, are likely influenced by the range of known viruses with available sequence data. These issues highlight the need for a careful choice of training dataset in modelling studies, but also for wider sampling and surveillance of coronaviruses among the wild virome [74], especially considering their high public health risks.

Although we focus on compositional features, predictive approaches using other sequence properties may improve more mechanistic understanding of host range. For example, amino acid composition and physicochemical similarity between contiguous amino acid residues can predict human origin of coronavirus spike sequences [34]. Similarly, Young et al. have recently demonstrated the use of multiple types of genomic features in combination to predict infected hosts and found physicochemical classification of amino acid k-mers to achieve similar predictive power to nucleotide k-mers [31]. More widely, hydrophobic and hydrophilic composition of host receptors shows some predictive signal towards virus sharing [75], hydropathy being of mechanistic importance during virus-receptor binding, e.g. for murine coronavirus [76]. These properties could be used as additional features and improve machine learning model-derived predictions.

This emphasises an additional key question for future modelling studies distinct from host origin—whether genomic traits can predict the zoonotic potential of newly discovered animal coronaviruses [77]. As this is strongly determined by molecular mechanisms of virus-host receptor interactions [22], these predictions may be best inferred by model frameworks combining genomic features of both spike proteins and host receptors.

## Conclusion

By training machine learning models on genome composition across the *Coronaviridae*, we demonstrate a detectable evolutionary signature predictive of host type rooted in a region of the genome that is key to host shifts. Our random forest predictions add to the growing evidence COVID-19 ultimately originated within bats, though further work is needed to understand the potential for intermediate hosts. Characterising spike proteins (and by extension, their interaction with host receptors) may provide a fruitful path to further understanding zoonosis risk among coronaviruses.

## Supporting information

**S1 Methods. Alternative sampling methodologies for random forest training datasets.**
(DOCX)

**S1 Data. Spike protein sequences used in analysis with metadata and model-predicted host category.** List of non-zoonotic coronavirus genome sequences used in spike protein machine learning analysis along with genus, GenBank accession ID, metadata-derived host category and predicted host category from random forest when respective coronavirus was held-out during validation. Coronavirus names refer to taxonomic entities as defined by NCBI taxonomy (whether species or unranked sub-species). Model predictions are given as single category with highest probability, along with probabilities associated with each individual category.
(CSV)

**S2 Data. Whole genome sequences used in analysis with metadata and model-predicted host category.** List of non-zoonotic coronavirus genome sequences used in whole genome machine learning analysis along with genus, GenBank accession ID, metadata-derived host category and predicted host category from random forest when respective coronavirus was held-out during validation. Coronavirus names refer to taxonomic entities as defined by NCBI taxonomy (whether species or unranked sub-species). Model predictions are given as single category with highest probability, along with probabilities associated with each individual category.
(CSV)

**S3 Data. Spike protein sequences of zoonotic human coronaviruses with metadata and model-predicted host category.** List of zoonotic coronavirus genome sequences sampled from humans used in spike protein machine learning prediction along with genus, GenBank accession ID, and predicted host category, aggregating over all random forests. Coronavirus names refer to taxonomic entities as defined by NCBI taxonomy (whether species or unranked sub-species). Model predictions are given as probabilities associated with each individual category.
(CSV)

**S4 Data. Whole genome sequences of zoonotic human coronaviruses with metadata and model-predicted host category.** List of zoonotic coronavirus genome sequences sampled from humans used in whole genome machine learning prediction along with genus, GenBank

accession ID, and predicted host category, aggregating over all random forests. Coronavirus names refer to taxonomic entities as defined by NCBI taxonomy (whether species or unranked sub-species). Model predictions are given as probabilities associated with each individual category.
(CSV)

**S1 Table. Distribution of sequences represented per coronavirus.** Descriptive statistics of number of sequences per coronavirus species or unranked subspecies (i.e., unique taxonomic ids) pre- and post-data thinning procedure to a maximum of 20 sequences per host-virus combination. Data shown separately for coronavirus spike proteins and whole genome sequences.
(DOCX)

**S2 Table. Number of coronavirus sequences represented per host category.** Number of genome sequences pre and post-data thinning procedure (to a maximum of 20 sequences per host-species combination) and number of coronavirus species or unranked subspecies (i.e., unique taxonomic ids) sourced from each host category. Data shown separately for coronavirus spike proteins and whole genome sequences. Note that although most coronaviruses were only known to infect a single host category, several coronaviruses infected multiple host categories and are represented across multiple counts of taxonomic ids.
(DOCX)

**S3 Table. Predictive performance of random forest models using spike protein predictor features applying alternative sampling methodologies.** Model diagnostics describing overall performance using spike protein predictor features as in Table 2 when applying alternative sampling methodologies for class imbalance. Each methodology was applied to training data, retaining the same outer validation sets in all cases for comparability. CI denotes confidence interval, Kappa denotes Cohen's Kappa statistic, mAUC denotes multiclass area-under-curve statistic, and F1macro denotes F1 score calculated using macro-averaging (performance on each host category weighted equally).
(DOCX)

**S4 Table. Predictive performance of random forest models using whole genome predictor features applying alternative sampling methodologies.** Model diagnostics describing overall performance using whole genome predictor features as in Table 2 when applying alternative sampling methodologies for class imbalance. Each methodology was applied to training data, retaining the same outer validation sets in all cases for comparability. CI denotes confidence interval, Kappa denotes Cohen's Kappa statistic, mAUC denotes multiclass area-under-curve statistic, and F1macro denotes F1 score calculated using macro-averaging (performance on each host category weighted equally).
(DOCX)

**S5 Table. Variance of predictive performance of random forest models over different random seeds.** Model diagnostics describing overall performance as in Table 2, repeating analyses for an additional ten random seeds. All metrics given represent the mean diagnostic with standard deviation in brackets. Kappa denotes Cohen's Kappa statistic, mAUC denotes multiclass area-under-curve statistic, and F1macro denotes F1 score calculated using macro-averaging (performance on each host category weighted equally).
(DOCX)

**S6 Table. Predictive performance of random forest models using spike protein predictor features, stratified by host category.** Model diagnostics describing overall performance when applied to predict host category of held-out coronaviruses. Balanced accuracy denotes $0.5^*$

(sensitivity + specificity).
(DOCX)

**S7 Table. Predictive performance of random forest models using whole genome predictor features, stratified by host category.** Model diagnostics describing overall performance when applied to predict host category of held-out coronaviruses. Balanced accuracy denotes 0.5*
(sensitivity + specificity).
(DOCX)

**S8 Table. Predictive performance of random forest models using envelope protein predictor features.** Model diagnostics describing overall performance using envelope protein predictor features as in Table 2. The same data thinning and cross-validation methodologies were applied as in Materials and Methods, thinning 1781 down to 458 envelope protein sequences from 144 coronaviruses. CI denotes confidence interval, Kappa denotes Cohen's Kappa statistic, mAUC denotes multiclass area-under-curve statistic, and F1macro denotes F1 score calculated using macro-averaging (performance on each host category weighted equally).
(DOCX)

**S1 Fig. Structured data partitioning and machine learning procedure.** Data partitioning diagram indicating the machine learning procedure used, distinguishing A) the outer loop using hold-one-out cross-validation applied to coronavirus species or unranked subspecies (with the aim of validating model performance) from B) the inner loop using 10-fold cross-validation (with the aim of optimising model parameters). Distribution of outcome classes (host category) were preserved when sampling data folds during each inner loop. Zoonotic coronavirus sequences sampled from humans that were not used for model training are also distinguished. RF denotes a random forest model, while 'spike' and 'wgs' refer to spike protein feature dataset and whole genome feature dataset, respectively.
(TIF)

**S2 Fig. Performance of random forests during parameter optimisation.** Performance of random forest models during inner loop of 10-fold cross-validation for parameter optimisation during A) spike protein and B) whole genome analysis. Y axis denotes prediction accuracy on test fold, X axis denotes minimum number of genome sequences in nodes at which tree algorithm continues to split, and colour denotes number of genomic features randomly considered at each splitting point. Boxes denote median and upper/lower quartiles, with whiskers extending to 1.5*IQR.
(TIF)

**S3 Fig. Dinucleotide biases by position within codon reading frames.** Calculated dinucleotide biases for coronavirus A) spike protein and B) whole genome sequences. Boxes denote median and upper/lower quartiles, with whiskers extending to 1.5*IQR. Colour denotes position of dinucleotide within codon reading frames, i.e., blue boxes denote dinucleotides spanning adjacent codons (position 3–1). Dashed grey line denotes null value of 1, indicating no difference in dinucleotide usage from expectation.
(TIF)

**S4 Fig. Random forest host predictions based on coronavirus genome composition, blocked by genus.** Stacked bar plots of predicted probabilities of each host category obtained from random forest models trained on A) spike protein and B) whole genome composition features as in Fig 3 when separated and ordered by genus (either *Alphacoronavirus*, *Betacoronavirus*, *Gammacoronavirus*, *Deltacoronavirus*, or "U" to indicated unassigned*)*, with

secondary ordering of largest to smallest probability of the correct host.
(TIF)

**S5 Fig. Random forest host predictions based on coronavirus genome composition, blocked by species or subspecies.** Stacked bar plots of predicted probabilities of each host category obtained from random forest models trained on A) spike protein and B) whole genome composition features as in Fig 3 when separated and ordered by species or unranked subspecies (i.e., unique taxonomic ids), with secondary ordering of largest to smallest probability of the correct host. See S1 Text for species key.
(TIF)

**S6 Fig. Partial dependence plots for most informative genomic features of spike proteins.** Model-predicted marginal probability of each coronavirus host category as functions of the four most informative genome composition bias features of spike protein sequences. A)–D) depict random forests trained on spike protein sequences and E)–H) depict probabilities as functions of the same features within random forests trained on whole genome sequences for comparison. Lines denote median values across A)–D) m = 225 and E)–H) m = 187 random forests during hold-one-out cross-validation. Shaded areas denote 2.5th and 97.5th percentiles. Colour key denotes host category.
(TIF)

**S7 Fig. Partial dependence plots for most informative genomic features of whole genome sequences.** Model-predicted marginal probability of each coronavirus host category as functions of the four most informative genome composition bias features of whole genome sequences. A)–D) depict random forests trained on whole genome sequences and E)–H) depict probabilities as functions of the same features within random forests trained on spike protein sequences for comparison. Lines denote median values across A)–D) m = 187 and E)–H) m = 225 random forests during hold-one-out cross-validation. Shaded areas denote 2.5th and 97.5th percentiles. Colour key denotes host category.
(TIF)

**S1 Text. Number key for coronavirus species or unranked subspecies as depicted in S5 Fig.**
(DOCX)

## Acknowledgments

We thank Cillian Courtney, Maya Wardeh, and Matthew Baylis for helpful support and commentary on this work.

## Author Contributions

**Conceptualization:** Liam Brierley, Anna Fowler.

**Data curation:** Liam Brierley.

**Formal analysis:** Liam Brierley.

**Funding acquisition:** Liam Brierley.

**Investigation:** Liam Brierley.

**Methodology:** Liam Brierley.

**Supervision:** Anna Fowler.

**Validation:** Liam Brierley.

**Visualization:** Liam Brierley.

**Writing – original draft:** Liam Brierley.

**Writing – review & editing:** Liam Brierley, Anna Fowler.

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
