## [Decision Letter · Decision Letter 0]

6 Jan 2021

Dear Brierley,

Thank you very much for submitting your manuscript "Predicting the animal hosts of coronaviruses from compositional biases of spike protein and whole genome sequences through machine learning" for consideration at PLOS Pathogens. As with all papers reviewed by the journal, your manuscript was reviewed by members of the editorial board and by several independent reviewers. In light of the reviews (below this email), we would like to invite the resubmission of a significantly-revised version that takes into account the reviewers' comments.

Thank you for your submission. Both reviewers appreciated your attention to an interesting topic. We would be happy to consider a revised manuscript. However, both reviewers raise a number of comments, which are not entirely overlapping. A revision would need to adequately address these comments and would likely be sent out for re-review.

We cannot make any decision about publication until we have seen the revised manuscript and your response to the reviewers' comments. Your revised manuscript is also likely to be sent to reviewers for further evaluation.

Sincerely,

Adam S. Lauring

Section Editor

PLOS Pathogens

Adam Lauring

Section Editor

PLOS Pathogens

Kasturi Haldar

Editor-in-Chief

PLOS Pathogens

orcid.org/0000-0001-5065-158X

Michael Malim

Editor-in-Chief

PLOS Pathogens

orcid.org/0000-0002-7699-2064

Thank you for your submission. Both reviewers appreciated your attention to an interesting topic. We would be happy to consider a revised manuscript. However, both reviewers raise a number of comments, which are not entirely overlapping. A revision would need to adequately address these comments and would likely be sent out for re-review.

Reviewer's Responses to Questions

**Part I - Summary**

Reviewer #1: The authors analyze spike protein and whole genomes of various coronaviruses using measures such as dinucleotide and codon usage biases to predict animal host using random forests. The approach appears to be surprisingly successful given the relatively simple measures used. Such an approach may be useful in determining genomic biases that are preferred by various coronaviruses as they adapt to their respective hosts.

Reviewer #2: The paper uses the compositional signatures of various coronaviruses to predict hosts with the aim of identifying the animal host of SARS-CoV-2. To do so, the authors use the random-forest machine-learning algorithm for classification, followed by analysis of the importance of each genomic feature.

While the development of algorithms for host prediction are extremely important (the current coronavirus emergence scenario is a good example), my general opinion is that such methods have to be transparent in their predictive mechanism. A black box approach, as used in the current study is potentially susceptible to a range of artefacts, of which over-fitting of compositional features through the underlying clustering of sequences by phylogeny relationships represents a substantial underlying compounding factor.

As I see it, (although I would be interested to hear the authors’ opinion on this), the basic problem is the compositional similarities used in the algorithm, such as codon choice and dinucleotides that specify amino acids, can also originate simply from virus sequences being similar to each other through genetic relatedness. Assignment of host based on compositional features might then simply represent a match to the phylogenetic clusters of viruses infecting that particular host. Indeed, although the figure is unlabelled, genus assignments seem to correlate with their host predictions in Fig. 2. Predicted hosts of pig coronaviruses (delta, alfa and beta), might correspond to the three blocks in host prediction. The alfa and bertacoronavirus blocks (on the right) furthermore look remarkably similar to the patterns of the human alfa and beta human seasonal human coronavirus. Similarly for camel viruses, where the first block might be 229E (Alfacoronavirus) and the second MERS-CoV-2-related (Betacoronavirus) based on the relative numbers of camel viruses provided in Table S1. At the very least the figure should be annotated to provide the genus assignments to demonstrate the extent to which these contribute to host prediction patterns.

In relation to that, I think it is a bit disingenuous to claim that MERS-CoV-2 was excluded from the training (lines 246-248) since the camel betacoronaviruses are very similar to MERS-CoV found in humans and it is therefore no surprise that the latter are then found to have a predicted camel origin (Fig. 3).

In terms of informative features, the observation that biases in codon positions 1 and 2 were the most predictive of host (with GG being the highest rank) might simply follow differences in amino acid composition between sequences that co-associated with genus.

**Part II – Major Issues: Key Experiments Required for Acceptance**

Reviewer #1: -The authors consider the spike protein as a special case but it may be that other proteins such as orf1ab or the RdRp may yield better results. At the very least some discussion should be added as to what would be expected if other gene(s) were analyzed. Also, a more detailed description of spike protein variation in the introduction would improve the justification for focusing on spike. The current description is fairly minimal.

-There are 116 genomic features used for the ML dataset. Can these be explicitly listed somewhere, e.g. in a supplementary table?

-The under-sampling strategy leads to a dramatic reduction in data utilized. Might it be possible to avoid this by instead introducing class weights during training?

-As far as host categories were concerned, did the model allow for the possibility of same virus but multiple hosts? Multiple bats, for example, may be able to harbor the same virus.

-The inner/outer loop cross validation strategy is unusual and makes it unclear whether the train and test sets are really independent. If they are not independent, the results may be marred by overfitting. How different are the results if only one or the other of these strategies (leave-one-out or 10-fold cross-validation) were used?

-The authors compare their "spike protein" and "whole genome" models at least twice: in the second paragraph of results ("Hierarchical clustering based on RSCU 182values suggested codon usage was less distinct between genera within spike protein sequences") and then in the discussion ("which may explain the stronger prediction of carnivore hosts when using spike protein sequences''). However, in neither case they provide any statistical support for their statements. Are these differences statistically significant? Similarly, the statement “Among our dataset of 225 coronaviruses, we observed A- and U-ending codons to be overrepresented and G- and C- ending codons to be underrepresented” in the discussion is not analyzed statistically.

Reviewer #2: 1. There is no published source code for this analysis, rendering it unable to be completely reproduced. At the minimum, all code used to generate the results and figures should be published.

2. The paper does not make clear the rationale behind using random forests for modelling the hosts of coronaviruses. Given the class imbalance was resolved with under-sampling, other statistical classification approaches such as support vector machines, logistic regression, and k-nearest neighbors may also be useful. Indeed, some of these approaches (such as k-NN) are robust to class imbalance in the first place and no explanation for their exclusion is provided in the paper. A fuller comparison of the methods and their relative performances would strengthen the methodological advances claimed in the manuscript.

3. The description of the training method is not fully clear and would be served by further explanation. For example, the reason the parameters were optimized within the inner loop is not fully explained within the main manuscript.

4. The paper is unclear whether dinucleotide frequencies, RSCU, and ENc were computed using a package and, if so, which package.

5. In relation to reproducibility, the manuscript does not describe whether a fixed random seed was used for the analysis and chosen beforehand. Doing so with the code supplied (see comment 1 above) would allow reproduction of the exact results presented as well as examination of the method over a variety of random seeds to ascertain its general performance.

6. Similarly, it is not explained in the paper how many times the analyses were run. In the absence of such information and reproducible source code, multiple testing can be a concern for the validity of the results.

7. The paper does not fully describe on the rationale for selecting twenty random samples per host category per virus for under-sampling. There are methods for under-sampling based on selecting nearest neighbors that can maintain or improve performance while also reducing class imbalance. Examples of such methods are near miss and condensed nearest neighbor under-sampling.

**Part III – Minor Issues: Editorial and Data Presentation Modifications**

Reviewer #1: -Typo on line 157: should read “rather than”.

Reviewer #2: Lines 36, 39. Why are these called intermediate hosts?

PLOS authors have the option to publish the peer review history of their article (what does this mean?). If published, this will include your full peer review and any attached files.

Reviewer #1: No

Reviewer #2: No
---

## [Decision Letter · Decision Letter 1]

9 Apr 2021

Dear Brierley,

We are pleased to inform you that your manuscript 'Predicting the animal hosts of coronaviruses from compositional biases of spike protein and whole genome sequences through machine learning' has been provisionally accepted for publication in PLOS Pathogens.

Best regards,

Adam S. Lauring

Section Editor

PLOS Pathogens

Adam Lauring

Section Editor

PLOS Pathogens

Kasturi Haldar

Editor-in-Chief

PLOS Pathogens

orcid.org/0000-0001-5065-158X

Michael Malim

Editor-in-Chief

PLOS Pathogens

orcid.org/0000-0002-7699-2064

Reviewer Comments (if any, and for reference):

Reviewer's Responses to Questions

**Part I - Summary**

Reviewer #1: The revised version is satisfactory, and the manuscript is ready for publication.

Reviewer #2: The authors have performed an excellent job of modifying the manuscript to address the reviewers' comments, in particular in their demonstration of effective host prediction independently of phylogenetic clustering. It's up to the authors, but in many ways, Fig. S4 is more informative in this respect than Fig. 2 (ie. showing moderately effective host prediction in each genus) and therefore removing one obvious source of compounding error with the approach. I would be happy see Fig. 2 updated if the authors wished.

**Part II – Major Issues: Key Experiments Required for Acceptance**

Reviewer #1: (No Response)

Reviewer #2: None

**Part III – Minor Issues: Editorial and Data Presentation Modifications**

Reviewer #1: (No Response)

Reviewer #2: See above - Fig. 2

PLOS authors have the option to publish the peer review history of their article (what does this mean?). If published, this will include your full peer review and any attached files.

Reviewer #1: No

Reviewer #2: No

---

## [Editor Report · Acceptance letter]

15 Apr 2021

Dear Brierley,

We are delighted to inform you that your manuscript, "Predicting the animal hosts of coronaviruses from compositional biases of spike protein and whole genome sequences through machine learning," has been formally accepted for publication in PLOS Pathogens.

Best regards,

Kasturi Haldar

Editor-in-Chief

PLOS Pathogens

orcid.org/0000-0001-5065-158X

Michael Malim

Editor-in-Chief

PLOS Pathogens

orcid.org/0000-0002-7699-2064